# Entropy-Argumentative Concept of Computational Phonetic Analysis of Speech Taking into Account Dialect and Individuality of Phonation

**DOI:** 10.3390/e24071006

**Published:** 2022-07-20

**Authors:** Viacheslav Kovtun, Oksana Kovtun, Andriy Semenov

**Affiliations:** 1Department of Computer Control Systems, Faculty of Intelligent Information Technologies and Automation, Vinnytsia National Technical University, Khmelnitske Shose Str., 95, 21000 Vinnytsia, Ukraine; kovtun_v_v@vntu.edu.ua; 2Department of the Theory and Practice of Translation, Faculty of Foreign Languages, Vasyl’ Stus Donetsk National University, 600-Richchya Str., 21, 21000 Vinnytsia, Ukraine; 3Department of Information Radioelectronic Technologies and Systems, Faculty of Information Electronic Systems, Vinnytsia National Technical University, Khmelnitske Shose Str., 95, 21000 Vinnytsia, Ukraine; semenov.a.o@vntu.edu.ua

**Keywords:** relative entropy, computational linguistics, computational phonetic analysis of speech, phonetic fusion, recognition of language units, individual phonetic alphabet

## Abstract

In this article, the concept (i.e., the mathematical model and methods) of computational phonetic analysis of speech with an analytical description of the phenomenon of phonetic fusion is proposed. In this concept, in contrast to the existing methods, the problem of multicriteria of the process of cognitive perception of speech by a person is strictly formally presented using the theoretical and analytical apparatus of information (entropy) theory, pattern recognition theory and acoustic theory of speech formation. The obtained concept allows for determining reliably the individual phonetic alphabet inherent in a person, taking into account their inherent dialect of speech and individual features of phonation, as well as detecting and correcting errors in the recognition of language units. The experiments prove the superiority of the proposed scientific result over such common Bayesian concepts of decision making using the Euclidean-type mismatch metric as a method of maximum likelihood and a method of an ideal observer. The analysis of the speech signal carried out in the metric based on the proposed concept allows, in particular, for establishing reliably the phonetic saturation of speech, which objectively characterizes the environment of speech signal propagation and its source.

## 1. Introduction

Computational phonetic analysis is a fundamental component of most information technologies for natural language recognition, cognitive speech analysis, automated speech transcription, and so on. The high reliability of phonetic analysis is a guarantee of a qualitative result of the functioning of all of these types of systems. The primary phonetic-morphological analysis of inflected languages and speech is especially relevant. The main source of errors in this process is a fusion [1,2,3,4]. This phenomenon characterizes the high variability of the individual sounding of phonemes, especially at the junction of morphemes. The phenomenon of fusion is objectively determined by the phonological evolution of natural language and cannot be ignored in the creation of precision technologies for computational phonetic analysis of speech.

The task of computational phonetic-morphological analysis of language or speech is objectively complicated, firstly, by the peculiarities of language itself as a process of physiologic-cognitive human activity, and, secondly, by the peculiarities of the profile information technologies involved.

We note the main integral factors of the first source of complications [5,6,7,8,9].

*Homonymy of inflections*. Inflexions can be homonymous if they belong to a single world-changing paradigm or characterize a single lexical and grammatical category, but belong to different world-changing paradigms, and they are sometimes found in the paradigms of different parts of language. This factor is a source of ambiguity in phonetic-morphological analysis. The negative impact of this factor can be reduced by using information technologies of linguistic context analysis and computational phonetic analysis.

*Internal flexion*. This type of inflexion is manifested when using the basic collection of language units, the representativeness of which depends on the content of word forms. If the collection is not used, it is necessary to formulate the rules of linguistic polymorphism inherent in the studied language.

*Complex lexemes*. Lexemes, the phonation (inscription) of which includes specific articulation techniques (special symbols), require the definition of declension for each component in the word form.

*Analytical word forms*. Analytical word forms are found in many languages and can cause significant complications in phonetic-morphological analysis because the components of the word form can be separated and even be located in different positions in the sentence.

*Large lexical fund of language*. Despite the rapid positive dynamics of computing power characteristics and the large memory capacity of modern computer technology, working with a basic collection of language units of the studied language (especially with a basic collection of word forms) in the implementation of phonetic-morphological analysis remains a task of high computing.

*Variability of the lexical level of language*. Updating the collections of language units for the appropriate type of phonetic-morphological analysis system does not keep up with the polymorphism of natural language (especially if we take into account dialects), which is manifested in the everyday phenomenon of new lexemes (specifically, terms) and word forms. Systems of computational phonetic-morphological analysis of a no collection type suffer less from the influence of this complicating factor.

We have mentioned only the most common factors of natural linguistic origin which negatively affect the effectiveness of computational phonetic-morphological analysis of speech. Depending on the information technology involved, this list is expanding.

We investigate the current state of the theoretical and analytical basis of current information technologies of computational phonetic-morphological analysis. Based on the results of information retrieval [10,11], we distinguish two relevant approaches—rationalistic and empirical. The first approach uses linguistic knowledge to analyze and synthesize language units. The second approach is based on the generalization of empirical data, for example, in the form of a statistical model of language (speech) [12,13]. However, in modern computational linguistics, technologies that integrate both of these approaches in a certain proportion are the most productive. According to the content of the used collection of language units, as a system-forming element for the implementation of computational phonetic-morphological analysis, technology-analyzers can be divided into [14,15,16,17]: (1) systems with a collection of phonemes and morphemes; (2) systems with a collection of lexemes and word forms; and (3) systems without basic collections.

The central element of the systems of the first type is a collection of relatively phonetically and linguistically stable language units (morphemes, phonemes, selected allophones) of the studied language. The corresponding technology analyzer decomposes the speech signal (text) into a certain sequence of indivisible portions, carrying out a recognition procedure for each of them. Such an elementary combinatorial model is most often used for the analysis of inflectional and agglutinative languages. The order of parts of lexemes is defined as the concatenation of the corresponding classes of morphemes in the collection. To determine the order of transition between classes of morphemes, the mathematical apparatus of finite state machines or Markov chains are usually used [18]. The number of classes of morphemes in the collection is determined by the result of the previous morphological classification of the studied language. In addition to declarative information on the composition of morphemes, the collection may also store procedural information. Such information determines the allowable range of variation of the patterns of morphemes and is most often designed as a system of production rules [10,11,14,19]. Empirical probabilistic-statistical methods are most often used for recognition in systems of this type [4,12,20].

Systems of the second type are focused on computational morphological analysis. Accordingly, the content of collections of etalons of language units in such systems is formed by morphemes and short lexemes. Systems of this type consider word forms as a sequence of such language units formed according to compositional and(or) production rules. When studying word forms, the system generates a lemma for it according to certain rules [21,22]. If such a lemma is present in the basic collection, then the word form is considered recognized. If we take into account the resource intensity, the effectiveness of such systems is determined mainly by the representativeness of the content of the basic collection. Collections of morphemes or lexemes are used in phonetic-morphological analysis to normalize the studied word forms. In the presence of a collection of morphemes, normalization is realized in the form of stemming. In the presence of a collection of lexemes, normalization is realized in the form of lemmas. We separately mention the subclass of systems of the second type, which uses a collection of word forms. The purpose of such systems is grammatical and morphological analysis, in which the collection presents a set of combinations of word forms, which is matched by a set of grammatical labels [11,19,23]. With a sufficiently rich collection, the source of analysis errors in these systems is only the homonymy of the complete word form.

The disadvantage of all systems of phonetic-morphological analysis of the first and second types is the use of large collections of language units. However, according to this criterion, systems focused on the use of phonetic and morphological collections look better if the efficiency of the recognition process is acceptable.

Systems of the third type perform phonetic-morphological analysis exclusively based on mathematical methods of machine learning (support vector machines, EM-method, genetic algorithms, Kohonen networks, etc.) [24,25,26,27,28,29,30,31,32,33,34]. Any methods capable of graphemic analysis [24], the result of which is the automatic or automated formation of phonetic-morphological collections, are acceptable. The advantage of the third type of system is the methodologically determined high heuristics and adaptability, which potentially allow for recognizing language units in speech material with a clear uncertainty. The disadvantage of such systems is the complexity and instability of learning these pseudo-intelligent methods, as well as the need for initial data and computing resources, the volume of which exceeds that required for systems of the first and second type, not in times, but orders.

Below we formulate the main provisions of our study.

The ***object*** of study is the fusion of the process of merged speech.

Considering the mentioned advantages and disadvantages of systems of phonetic-linguistic analysis, we formulate the ***purpose*** of the study as formalization in the paradigm of information theory of a statistically adequate analytically rigorous concept of phonetic analysis of speech, the variability of which will be taken into account.

The ***subject*** of research will be methods of probability theory and mathematical statistics, information theory, pattern recognition theory and acoustic theory of language formation.

In this context, the ***objectives*** of the study are: to create a concept of the process of computational phonetic analysis of speech, taking into account dialects and the specifics of phonation introduced by the speaker; to formulate a criterion for the estimation of the phonetic saturation of speech based on the proposed model, taking into account the distorting effect of the channel of propagation of speech signals in the phonation process; and to prove the adequacy and functionality of the obtained theoretical results.

The ***main contribution*** of the research is the concept of computational phonetic analysis of speech. In the concept, in contrast to the existing methods, the task of addressing the multicriteria of the process of cognitive perception of speech by a person is strictly formally presented in the theoretical and analytical apparatus of information theory, pattern recognition theory and acoustic theory of speech formation. The obtained concept allows for determining accurately the phonetic alphabet of a person, taking into account their inherent dialect of speech and individual features of phonation, as well as detecting and correcting errors in the recognition of language units and reliably assessing the phonetic saturation of speech.

The ***highlights*** of this research are:-The entropy-argumentative concept (i.e., the mathematical model and methods) of computational phonetic analysis of speech, taking into account dialect and individuality of phonation;-The entropy-argumentative concept of detection and correction of errors of computational phonetic analysis of speech.

## 2. Materials and Methods

### 2.1. Statement of Research

The functional purpose of typical modern information technology of computational analysis of speech patterns is realized by comparing the parameterized representation of the studied language unit and its corresponding etalon in a certain parametric space. The main source of uncertainty in the comparison process is the biological origin of the speech signal and its distortion during transmission and processing. However, the acoustic variability of phonation of language units (primarily, phonemes), due to the existence of dialects, is relatively stable. Based on this fact, we assume a simultaneous comparison of the studied pattern of the phonogram with the pronounced phoneme x with each element $x_{r,j}$ of the set of etalons $X_{r}=\left\{x_{r,j}\right\}$, where $j=\bar{1,J_{r}}$ is the index of the etalon that characterizes the corresponding dialect of the phoneme $r=\bar{1,R}$, where R is the capacity of the phonetic alphabet and $J_{r}$ is the capacity of the set of recognized dialects for the phoneme r. Then, if the distance $\rho \left(x/x_{r,j}\right)$, $r=\bar{1,J_{r}}$, between the studied pattern x and at least one of the elements $x_{r,j}$ of the cluster of the r-th phoneme does not exceed the specified threshold value
(1)$$\frac{1}{J_{r}}\sum_{j=1}^{J_{r}}\rho \left(\frac{x}{x_{r,j}}\right)\leq \rho_{0},$$
then we can recognize the pattern x as the phoneme $r\in X_{r}$. Such a process of recognizing language units will be objective (in particular, insensitive to the dialects of phonation of language units), as the clusters $\left\{x_{r,j}\right\}$ for the phonetic alphabet $X_{r}$ are representatively defined. Depending on the value of the threshold $\rho_{0}$, the result of the analysis of the studied pattern x according to Rule (1) will be: its recognition as one of the phonemes: x=r; its identification with several phonemes: $x=\left\{r_{i}\right\}$, $r_{i}\in X_{r}$, $i\leq J_{r}$; or its recognition as marginal regarding the studied phonetic alphabet: $x\neq \forall r\in X_{r}$. To simplify the calculations, we convert Rule (1) into the form
(2)$$\rho_{r}\left(x\right)=x_{r}^{*}=x_{r,v}:\frac{1}{J_{r}}\sum_{j=1}^{J_{r}}\rho \left(\frac{x_{r,j}}{x_{r,v}}\right)=\underset{i\leq J_{r}}{\min}\frac{1}{J_{r}}\sum_{j=1}^{J_{r}}\rho \left(\frac{x_{r,j}}{x_{r,i}}\right)≜\rho_{r}^{*}\leq \rho_{0},$$
where in the process of recognizing the pattern x within the cluster $X_{r}$ one distance $\rho_{r}\left(x\right)≜\rho \left(x/x_{r}^{*}\right)$ from it to the center of the cluster $x_{r}^{*}$ is calculated, the coordinates of which determine the dialect-averaged phoneme etalon $r\in X_{r}$.

Based on Rule (2), we define the procedure of computational phonetic analysis of speech as a comparison of empirical (spoken by the person) $\left\{x_{v}^{*}\right\}$ and etalon $\left\{x_{r}^{*}\right\}$ sets of equal capacity, the pairwise elements of which generalize the corresponding phonemes of the studied language both on the speaker’s side v∈V and on the side of the etalon phonetic collection r∈R.

### 2.2. Entropy-Argumentative Concept of Computational Phonetic Analysis of Speech Taking into Account Dialect and Individuality of Phonation

Based on the provisions of information theory, we argue the solution rule (2) in the context of the relative entropy functional [35,36,37] (3):(3)$$\rho \left(x\right)≜\int \ldots \int \ln\frac{dP\left(x\right)}{dP_{r}\left(x\right)}P\left(dx\right),$$
where $P\left(x\right)$ is the selective probability distribution of the studied (empirical) speech signal x relative to the etalon probability distribution $P_{r}\left(x\right)$, $r=\bar{1,R}$. Assume that the distribution law $P\left(x\right)$ is normal: $P\left(x\right)=N\left(K_{X}\right)$, where $K_{X}$ is a sample matrix of autocorrelation of the speech signal x of dimension n×n. Consider this in Expression (3): $\rho_{r}\left(x\right)=\frac{1}{2}\left(\mathrm{tr}\left(\frac{K_{x}}{K_{r}}\right)-\ln\left(\frac{K_{x}}{K_{r}}\right)-n\right)$, where $\mathrm{tr}\left(A\right)$ is the operation of finding a trace of the matrix A. If we assume that the studied speech signal is normalized to its entropy, then the last expression can be further simplified to the form
$$\rho_{r}\left(x\right)=\frac{1}{2}\left(\mathrm{tr}\left(\frac{K_{x}}{K_{r}}\right)-n\right).$$

We present Function (3) in frequency space as the optimal solving statistics [35]. For one sample of the studied speech signal, we obtain (4):(4)$$\rho_{r}\left(x\right)=\frac{1}{F}{\left|\frac{1-\sum_{m=1}^{p}a_{r}\left(m\right)e^{-j\pi m\frac{f}{F}}}{1-\sum_{m=1}^{p}a_{x}\left(m\right)e^{-j\pi m\frac{f}{F}}}\right|}^{2},$$
where f is the discrete frequency value for the analyzed sample of the speech signal, F is the upper limit value of the speech signal frequency equal to half of its sampling frequency, and $\left\{a_{r}\left(m\right)\right\}$ and $\left\{a_{x}\left(m\right)\right\}$ are the vectors of linear autoregression coefficients of order p for etalon signal $x_{r}^{*}$ and empirical signal x, respectively. The expression in the numerator of (4) is an amplitude-frequency characteristic of the bleaching filter tuned to highlight the features of the r-th phoneme $x_{r}^{*}$, $r=\bar{1,R}$.

Expressions (2) and (4) allow us to calculate quantitative characteristics, based on which it is possible to reasonably decide whether the studied pattern x belongs to the cluster $x_{r}^{*}$ of the corresponding phoneme $r\in X_{r}$. It is possible to vary the errors of this recognition process by changing the value of the threshold $\rho_{0}$. Given the Gaussian approximation of the speech signal, the probability of error of the first kind α for the process of phoneme recognition taking into account the dialects of the studied language is proposed to be defined in terms of $\chi^{2}$-criterion with M degrees of freedom:(5)$$\alpha ≜P\left\{{\left.\rho_{r}\left(x\right)\geq \rho_{0}\right|}_{x\in X_{r}}\right\}=P\left\{\chi_{M}^{2}>M\left(1+\rho_{0}\right)\right\},$$
where $P\left\{.\right\}$ is the probability of a random event, M=const.

In the general case, the value of the constant M is calculated by the expression M≈L−p, where p is the order of the bleaching filter, and L=2Fτ is a parameter whose value depends on the number of stationary intervals τ allocated in the studied speech signal x. The value of error α determined by Expression (5) is inversely proportional to the value of the threshold $\rho_{0}$. For example, for a given value of α=0.1 at τ=5 ms, F=8 kHz, p=20, we obtain L=80 and, accordingly, M=60. Using the $\chi^{2}$-distribution tables for the significance level β=1−α=1−0.99=0.01, we find the value of the quantile $\chi_{M;\beta }^{2}=\chi_{60;0.01}^{2}=88.38$, using which we calculate the value of the threshold $\rho_{0}$: $\rho_{0}=\chi_{M;\beta }^{2}/M-1=0.473$.

The error of the second kind β in the context of the task of computational phonetic analysis of speech when taking into account dialects represents the probability of the confusion of phonemes r and v, $r,v\in X_{r}$, the centers of clusters $x_{r}^{*}$ and $x_{v}^{*}$ of which are close enough in the parametric space $\rho_{rv}≜{\left.\rho_{r}\left(x\right)\right|}_{x=x_{v}^{*}}$. Therefore, the value of error β is inversely proportional to the value of distance $\rho_{rv}$. Analysis of the results of a statistically representative number of experiments showed that the minimum value of $\rho_{rv}$ the phonetic alphabets of the English language $\left\{x_{r}^{*}\right\}$ is in the range $\left[0.2;0.3\right]$. Accordingly, in analogy with (5), we formalize the expression for calculating the error of the second kind β of the phoneme recognition process taking into account the dialects of the studied language:(6)$$\beta ≜P\left\{{\left.\rho_{r}\left(x\right)\geq \rho_{0}\right|}_{x\in X_{v}}\right\}=P\left\{\chi_{M}^{2}<\frac{M\left(1+\rho_{0}\right)}{1+\rho_{rv}}\right\}.$$

Summarizing the considerations embodied in Expressions (5) and (6), for practical use we choose the value of the threshold $\rho_{0}$ in the decision rule (2) based on the expression
(7)$$p_{0}=\left(1,\ldots ,2\right)\underset{r,v}{\min\rho_{rv}}.$$

The value of the threshold $\rho_{0}$, calculated by Expression (7), provides a balance between the values of errors of the first and second kind of the process of phoneme recognition from the phonetic alphabet $X_{r}$, taking into account the dialects of the studied language and the variability of the phonation process. However, the question of the influence of individual features of speakers’ articulation on the result of phonetic analysis of speech requires more detailed analytical formalization.

In the context of the provisions of information theory, we consider the speaker as a source of discrete messages X, defined on the set of etalons of language units $\left\{x_{r}^{*}\right\}$. Such a source can be comprehensively characterized by the amount of information per language unit generated by it. 

If we ignore the influence of individual features of the speaker’s articulatory apparatus on the phonation process and assume that the speech message is transmitted in the absence of acoustic ambient noise, the required amount of information is defined as Shannon entropy for a discrete message source [35]:(8)$$H\left(X\right)≜-\sum_{r=1}^{R}P\left(X=x_{r}^{*}\right)\log P\left(X=x_{r}^{*}\right)=-\sum_{r=1}^{R}p_{r}\log p_{r}.$$

If we mention the normalization $\sum_{r=1}^{R}p_{r}=1$, then, considering the equally probable appearance of language units ∀r≤R: $p_{r}=1/R$, we obtain a simplified form of Expression (8): $H\left(X\right)=\log R$. However, in real conditions, it is impossible to ignore articulatory conditioned variability of phonation. The speech signal at the output of the articulatory tract of the speaker $X^{\prime }$ may differ significantly from the etalon X: $X^{\prime }\neq X$.

This axiom is true even for individual phonemes, not to mention more massive language units. Under such conditions, an adequate mathematical model of a discrete source of speech messages should be created based on phonemes defined by Expression (5), clearly clustered in the parametric space: $q_{r}≜P\left(X^{\prime }\neq x_{r}^{*}\right)$, $r=\bar{1,R}$, and taking into account the probability of an abstract, *R*+1-th, language unit, which includes cases of the unreliable recognition of a signal $X^{\prime }$: $q_{R+1}≜P\left(X^{\prime }\neq x_{r}^{*},\forall r\leq R\right)$. We summarize these considerations for the decision rule (2):(9)$$\begin{array}{c} q_{r}=\sum_{v=1}^{R}q_{rv}=\sum_{v=1}^{R}P\left(X^{\prime }=x_{r}^{*};X=x_{v}^{*}\right)=\sum_{v=1}^{R}P\left(X=x_{v}^{*}\right)P\left(X^{\prime }=x_{r}^{*}\left|X=x_{v}^{*}\right.\right)\\ =P\left(X=x_{r}^{*}\right)P\left(X^{\prime }=x_{r}^{*}\left|X=x_{r}^{*}\right.\right)=\left(1-\alpha \right)p_{r},\\ q_{R+1}=\sum_{v=1}^{R}P\left(X^{\prime }=x_{v}^{*};X=x_{v}^{*}\right)=\sum_{v=1}^{R}P\left(X=x_{v}^{*}\right)P\left(X^{\prime }\neq x_{v}^{*}\left|X=x_{v}^{*}\right.\right)=\sum_{v=1}^{R}\alpha p_{v}=\alpha ,\\ \sum_{r=1}^{R+1}q_{r}=\left(1-\alpha \right)\sum_{r=1}^{R}p_{r}+\alpha \equiv 1, \end{array}$$
where $P\left(X^{\prime }=x_{r}^{*}\left|X=x_{r}^{*}\right.\right)=1-\alpha$ is the conditional probability of recognizing the *r*-th phoneme, provided that the variability of its phonation introduced by the speaker is ignored.

Note that Expression (8) characterizes a discrete source of speech messages without taking into account the disturbing effect of the channel of their distribution on the final result of phonation. Consider this information using as a basic expression [35]:(10)$$I\left(X,X^{\prime }\right)≜H\left(X\right)-H\left(X\left|X^{\prime }\right.\right),$$
where X is a specimen of the phonation of the etalon $x_{r}^{*}$ of the phoneme $r\in X_{r}$, $X^{\prime }$ is a specimen of the phonation of this phoneme by the speaker (empirical specimen), and $H\left(X\left|X^{\prime }\right.\right)$ is a posteriori entropy, which characterizes the scattering of useful information of a phonation process due to disturbing effects in its distribution channel. Taking into account Expression (9), we formulate the equivalent representation of Expression (10):(11)$$\begin{array}{c} I\left(X,X^{\prime }\right)=H\left(X\right)+H\left(X^{\prime }\right)-H\left(XX^{\prime }\right)=H\left(X\right)-\sum_{r=1}^{R+1}q_{r}\log q_{r}+\sum_{v=1}^{R}\sum_{r=1}^{R+1}q_{rv}\log q_{rv}=H\left(X\right)\\ -\left(1-\alpha \right)\sum_{r=1}^{R}p_{r}\log\left(p_{r}\left(1-\alpha \right)\right)-\alpha \log\alpha +\sum_{r=1}^{R}q_{rr}\log q_{rr}+\alpha \sum_{v=1}^{R}p_{v}\log\left(p_{v}\alpha \right)=H\left(X\right)\\ +\left(1-\alpha \right)H\left(X\right)-\left(1-\alpha \right)\left(\left(1-\alpha \right)-\alpha \log\alpha +\left(1-\alpha \right)\sum_{r=1}^{R}p_{r}\log\left(p_{r}\left(1-\alpha \right)\right)\right)-\alpha H\left(X\right)\\ +\alpha \log\alpha =\left(1-\alpha \right)H\left(X\right). \end{array}$$

Based on Expression (11), we can say that the a posteriori entropy of information scattering in the phonation of the speech message $H\left(X\left|X^{\prime }\right.\right)$ is in direct proportion to the entropy of the discrete speech message source (8):(12)$$H\left(X\left|X^{\prime }\right.\right)=\alpha H\left(X\right).$$

Based on Expression (12), we can say that with an equally probable distribution of phonemes in the phonetic alphabet of the speaker, the upper limit of scattering of useful information in the phonation process can be described by the expression
(13)$$\sup H\left(X\left|X^{\prime }\right.\right)=\alpha \log R.$$

The obtained result correlates with the known Fano inequality [38] for arbitrary solution rules:(14)$$H\left(X\left|X^{\prime }\right.\right)\leq -\alpha \log\alpha -\beta \log\beta +\alpha \log\left(R-1\right).$$

The last statement can be proved empirically by comparing the calculated values of the right-hand sides of Expressions (13) and (14) for the experimental data for 0≤α≤1 and 1<R<∞.

Thus, the decision rule (2), the decision statistic (4) and Expressions (7)–(9) together form the desired concept of the process of computational phonetic analysis of speech, taking into account dialects and the specifics of phonation introduced by the speaker. The central element of the concept is the matrix of information mismatch $\left\|\rho_{r,v}\right\|$ of dimensions R×R. The data from the matrix $\left\|\rho_{r,v}\right\|$ are the basis for calculating the threshold $\rho_{0}$ using Expression (7). With a known value of $\rho_{0}$ based on Expressions (2) and (5), the procedure of segmentation of the phonetic alphabet $X_{r}=\left\{x_{r,j}\right\}$ into a set of phonemes, which with probability β=1−α are reliably recognized despite the above-described disturbing factors, and another set of phonemes, which with probability α are not reliably recognized. A significant factor for such segmentation is the probability of error of the first kind, which is calculated by Expression (5). The probability of error of the second kind (6) in this procedure is taken into account indirectly as a limitation in determining the threshold $\rho_{0}$ by Expression (7). The use of Expressions (9) and (10) allows for clarifying the result of the segmentation procedure, taking into account the variability of the phonation of the studied language units caused by the individual features of the articulation of a particular speaker. Note that although the presented concept was formulated based on phonemes, the provisions underlying it are consistent and for the analysis of speech about the content of such language units as morphemes and lexemes. Based on the proposed concept (8)–(10), Rule (11) allows us to estimate the error of the first kind (5) and the personalized entropy of the phonetic dictionary (8) as a result of the analysis of empirical data, the sample size of which is N=2FT. The statistically representative volume $N=10^{6}$ in the study of the phonetic alphabet of $R=10^{2}$ elements by Rule (11) as a result of analysis of phonograms of speech signals with a sampling frequency of 16 kHz is achieved with a censored duration.

### 2.3. Entropy-Argumentative Concept of Detection and Correction of Errors of Computational Phonetic Analysis of Speech

Let $X_{r}=\left\{x_{r,j}\right\}$, $r=\bar{1,R}$, $j=\bar{1,M}$ be a set of independent classified samples of type $x_{r,j}={\left[x_{r,j^{\left(1\right)}},x_{r,j^{\left(2\right)}},\ldots ,x_{r,j^{\left(n\right)}}\right]}^{T}$ with a capacity n of R≥2 Gaussian distributions $P_{r}=N\left(K_{r}\right)$ with zero mathematical expectation and unknown autocorrelation matrix $K_{r}={Ε}_{X}\left(x_{r,j}x_{r,j}^{T}\right)$ of dimension n×n, where j is the identifier of the cycle of observations of the r-th distribution, T is the transposition operation, $E_{X}$ is the mathematical expectation of the sample of sets X. Denote by $X_{0}$ a sample of the form $X_{r}$ with capacity $M_{0}$ for the studied signal with an unknown distribution $P\left(X\right)\subset \left\{P_{r}\right\}$. The task of recognizing the signal $X_{0}$ involves R-alternative testing of statistical hypotheses $W_{r}$ regarding the distribution law of this signal:(15)$$W_{r}:P\left(X\right)=P_{r},\;r=\bar{1,R}.$$

Let R=2, i.e., two competing hypotheses, $W_{1}:P\left(X\right)=P_{1}$ and $W_{2}:P\left(X\right)=P_{2}$, are tested for a priori unknown autocorrelation matrices $K_{1}$ and $K_{2}$. The verification will be performed using the asymptotic minimax criterion of the likelihood ratio [35,36,37] based on data from a sample $X\left\{X_{i}\right\}$, $i=\bar{0,2}$. Under such conditions, the hypothesis $W_{1}$ will be considered true if the condition
(16)$$W_{1}:\lambda_{1}\left(X\right)≜\frac{\underset{K_{1}}{\sup}\underset{K_{2}}{\sup}\left(p\left(X\left|W_{1}\right.\right)\right)}{\frac{\underset{K_{1}}{\sup}\underset{K_{2}}{\sup}\left(p\left(X\left|W_{2}\right.\right)\right)}{}}\equiv \frac{\underset{K_{1}}{\sup}\left(p\left(X_{0}\left|W_{1}\right.\right)p\left(X_{1}\right)\right)\underset{K_{2}}{\sup}\left(p\left(X_{2}\right)\right)}{\frac{\underset{K_{1}}{\sup}\left(p\left(X_{0}\left|W_{2}\right.\right)p\left(X_{1}\right)\right)\underset{K_{2}}{\sup}\left(p\left(X_{1}\right)\right)}{}}>1,$$
is satisfied, where $p\left(X_{0}\left|W_{r}\right.\right)$ is the plausibility function of the signal $X_{0}$ provided that hypothesis $W_{r}$ is confirmed, and $p\left(X_{r}\right)$ is the plausibility function of the signal $X_{r}$.

Using the known computational algorithm [38] under the condition of independence of observations $X_{r}=\left\{x_{r,j}\right\}$, we write a system of equations of the form
(17)$$\left\{\begin{array}{l} \ln\left(p\left(X_{0}\left|W_{r}\right.\right)\right)=-\frac{M_{0}}{2}\left(\ln\left|K_{r}\right|+\mathrm{tr}\left(\frac{S_{0}}{K_{r}}\right)+n\ln\left(2\pi \right)\right),\\ \ln\left(p\left(X_{r}\right)\right)=-\frac{M_{r}}{2}\left(\ln\left|K_{r}\right|+\mathrm{tr}\left(\frac{S_{r}}{K_{r}}\right)+n\ln\left(2\pi \right)\right), \end{array}\right.$$
where $\left|K_{r}\right|$ is the determinant of the matrix $K_{r}$, and $S_{r}≜\frac{1}{M_{r}}\sum_{j=1}^{M_{r}}x_{r,j}x_{r,j}^{T}$ is the estimate of the maximum likelihood for the matrix $K_{r}$ determined on the sample $X_{r}$, $r=\bar{0,2}$. We describe based on Rxpression (17) the fact that the upper limits $\ln\left(p\left(X_{r}\right)\right)$ are reached at $K_{r}=S_{r}$:(18)$$\underset{K_{r}}{\sup}\left(p\left(X_{r}\right)\right)=-\frac{M}{2}\left(\ln\left|S_{r}\right|+nc\right),$$
where $r=\left\{1,2\right\}$, $c=\ln\left(2\pi \right)+1$.

Similarly, we obtain the expression for determining the upper limits for $\ln\left(p\left(X_{0}\left|W_{r}\right.\right)p\left(X_{r}\right)\right)$:(19)$$\begin{array}{c} \underset{K_{r}}{\sup}\left(\ln p\left(X_{0}\left|W_{r}\right.\right)p\left(X_{r}\right)\right)\\ =-\frac{1}{2}\left(\left(M_{0}+M\right)\left(\ln\left|S_{0r}+n\ln\left(2\pi \right)\right|\right)+M_{0}\mathrm{tr}\left(\frac{S_{0}}{S_{0r}}\right)+M\mathrm{tr}\left(\frac{S_{0}}{S_{0r}}\right)\right)\\ =-\frac{M_{0}+M}{2}\left(\ln\left|S_{0r}\right|+nc\right), \end{array}$$
where $r=\left\{1,2\right\}$, and $S_{0r}=\frac{M_{0}}{M_{0}+M}\left(S_{0}+S_{r}\right)$ is the estimate of the maximum likelihood for the matrix $K_{r}$ determined on the combined sample $X_{0r}+\left\{X_{0},X_{r}\right\}$ with capacity $M_{0}+M$. 

Substitute Expressions (18) and (19) into Expression (16) and obtain the condition under which the hypothesis $W_{1}$ will be considered correct:(20)$$\begin{array}{c} W_{1}\left(X\right):\lambda_{1}\left(X\right)=\frac{1}{2}\left(\left(M_{0}+M\right)\ln\left|S_{01}\right|-\left(M_{0}-M\right)\ln\left|S_{02}\right|-M\ln\left|S_{1}\right|+M\ln\left|S_{2}\right|\right)<0\\ \equiv M_{0}\gamma_{1,01}+M\gamma_{1,01}<M_{0}\gamma_{2,02}+M\gamma_{2,02}, \end{array}$$
where $\gamma_{k,0r}=\frac{1}{2}\left(\mathrm{tr}\left(\frac{S_{k}}{S_{0r}}\right)-\ln\left|S_{k}\right|+\ln\left|S_{0r}\right|-n\right)\geq 0$ is the value of the relative entropy functional between two hypothetical probability distributions with autocorrelation matrices $S_{k}$ and $S_{0r}$.

We scale rule (20) for the task of recognizing signals of the form in (15) with an arbitrary number of hypotheses R≥2:(21)$$W_{v}\left(X\right):{\left.\left(M_{0}\gamma_{0,0r}+M\gamma_{r,0r}\right)\right|}_{r=v}=\min,\;r=\bar{1,R}.$$

Assuming the homogeneity of the pair of signals $X_{0}$ and $X_{r}$ in the sample $X_{0r}$ and considering that $\gamma_{0,0r}\leq \gamma_{0,r}$, $\gamma_{r,0r}\leq \gamma_{r,0}$ and $M=M_{0}$, we present Rule (21) in the form
(22)$$W_{v}\left(X\right):\lambda_{v}\left(X\right)≜{\left.\left(M_{0}\gamma_{0,r}+M\gamma_{r,0}\right)\right|}_{r=v}≜\left.\gamma_{0,r}+\gamma_{r,0}\right|r=v=\min,\;r=\bar{1,R}.$$
where the solving statistics of the relative entropy functional
(23)$$\gamma_{0,r}=\frac{1}{2}\left(\mathrm{tr}\left(\frac{S_{0}}{S_{r}}\right)-\ln\left|S_{0}\right|+\ln\left|S_{r}\right|-n\right),$$
(24)$$\gamma_{r,0}=\frac{1}{2}\left(\mathrm{tr}\left(\frac{S_{r}}{S_{0}}\right)-\ln\left|S_{r}\right|+\ln\left|S_{0}\right|-n\right)$$
are determined on the R-set of pairs of sample distributions $N\left(S_{0}\right)$, $N\left(S_{r}\right)$, $r=\bar{1,R}$. 

An alternative to Expressions (23) and (24) may be to take into account the principle of the minimum value of information non-directional mismatch $J\left(X_{0},X_{r}\right)≜\frac{1}{2}\left(\gamma_{0,r}+\gamma_{r,0}\right)$ between stochastic signals $X_{0}$ and $X_{r}$, $r=\bar{1,R}$, in the rule (22):(25)$${\tilde{W}}_{v}\left(X\right):{\tilde{\lambda }}_{v}\left(X\right)≜{\left.\gamma_{0,r}\right|}_{r=v}=\min,\;r=\bar{1,R},$$
where the decision statistics $\gamma_{0,r}$ are determined by Expression (23).

Expression (25) is a particular case of Criterion (22), provided that with an unlimited increase in the volume of training samples M, the second term in Expression (21) asymptotically reduces to zero: $\gamma_{r,0r}\to \gamma_{r,r}=0\forall r\leq R$. Thus, the transition from Rule (22) to (25) is appropriate provided that there is a significant asymmetry in the values of the decision statistics (23), (24).

The probability $\alpha_{v\to r}≜P\left(\left.W_{r}\left(X\right)\right|W_{r}\right)$ of confusion of the v-th and r-th signals, v≠r≤R, from the user database of a priori data $\left\{X_{r}\right\}$ in the formalism of Rule (22) can be described by the expression
(26)$$\alpha_{v\to r}=P\left\{\gamma_{0,v}+\gamma_{v,0}>\gamma_{0,r}+\gamma_{r,0}\left|W_{v}\right.\right\}=P\left\{2\gamma_{v,v}>\gamma_{v,r}+\gamma_{r,v}\right\}.$$

If we take into account that the empirical signal before recognition is normalized to the value of its specific entropy, then the system of asymptotic equations $\forall r\leq R:\frac{1}{n}\ln\left|S_{r}\right|=\frac{1}{n}\ln\left|S_{0}\right|\underset{n\to \infty }{=}\ln\sigma_{0}^{2}=const$ is satisfied. We take this fact into account by presenting the solving statistics $\gamma_{v,r}$ in the $\chi^{2}$-distribution formalism with K≤M degrees of freedom: $\gamma_{v,r}=\frac{1}{2}n\left(\frac{\sigma_{r,v}^{2}\sigma_{0}^{2}\chi_{r,v}^{2}\left(K\right)}{M}-1\right)$, where $\sigma_{r,v}^{2}≜\frac{\sigma_{0}^{2}}{n}\underset{n\to \infty }{\lim}\left(M\mathrm{tr}\left(\frac{S_{v}}{S_{r}}\right)\right)$ is an auxiliary variable. Substitute the obtained expression for statistics $\gamma_{v,r}$ into Expression (26):(27)$$\alpha_{v\to r}=P\left\{\sigma_{0}^{2}\chi_{v,v}^{2}>\frac{1}{2}\sigma_{r,v}^{2}\chi_{r,v}^{2}+\frac{1}{2}\sigma_{v,r}^{2}\chi_{v,r}^{2}\right\}=P\left\{2\chi_{v,v}^{2}>\left(1+\rho_{r,v}\right)\chi_{r,v}^{2}+\left(1+\rho_{v,r}\right)\chi_{v,r}^{2}\right\},$$
where $\rho_{r,v}≜\frac{\sigma_{r,v}^{2}}{\sigma_{0}^{2}}-1$ and $\rho_{v,r}≜\frac{\sigma_{v,r}^{2}}{\sigma_{0}^{2}}-1$ are the specific values of the information discrepancy for the studied pair of distributions $N\left(S_{0}\right)$ and $N\left(S_{r}\right)$ at n→∞, and $\sigma_{v,r}^{2}≜\frac{\sigma_{0}^{2}}{n}\underset{n\to \infty }{\lim}\left(M\mathrm{tr}\left(\frac{S_{r}}{S_{v}}\right)\right)$ is an auxiliary variable of the same type as $\sigma_{r,v}^{2}$. If we assume the mutual noncorrelation of the three $\chi^{2}$-distributions in Expression (27), then Expression (26) for calculating the probability of confusion $\alpha_{v\to r}$ can be represented as $\alpha_{v\to r}=P\left\{\frac{1}{2}\left(\left(1+\rho_{r,v}\right)F_{r,v}\left(1,K\right)+\left(1+\rho_{v,r}\right)F_{v,r}\left(1,K\right)\right)<1\right\}$, where $F_{r,v}\left(1,K\right)=\frac{\chi_{r,v}^{2}}{\chi_{v,v}^{2}}$ and $F_{v,r}\left(1,K\right)=\frac{\chi_{v,r}^{2}}{\chi_{v,v}^{2}}$ are statistics of the F-distribution with $\left(1,K\right)$ degrees of freedom. Accordingly, the upper limit of the probability of confusion $\alpha_{v\to r}$ can be estimated by the expression
(28)$$\begin{array}{c} \alpha_{v\to r}\leq P\left\{\frac{1}{2}\max\left[\left(1+\rho_{v.r}\right)F_{v,r}\left(1,K\right);\left(1+\rho_{r.v}\right)F_{r,v}\left(1,K\right)\right]\right\}\\ =P\left\{F\left(1,K\right)<\frac{2}{\max\left[\left(1+\rho_{v.r}\right);\left(1+\rho_{r.v}\right)\right]}\right\}=\\ P\left\{F\left(K,1\right)\geq \frac{1}{2}\max\left[\left(1+\rho_{v.r}\right);\left(1+\rho_{r.v}\right)\right]\right\}=1-\Phi_{K,1}\left\{\max\left[\left(1+\rho_{v.r}\right);\left(1+\rho_{r.v}\right)\right]\right\}, \end{array}$$
where $F\left(1,K\right)=\max\left[F_{r,v}\left(1,K\right);F_{v,r}\left(1,K\right)\right]$ and $F\left(K,1\right)=\frac{1}{F\left(1,K\right)}$ are statistics of the F-distribution with $\left(1,K\right)$ and $\left(K,1\right)$ degrees of freedom, respectively; $\Phi_{K,1}$ is the integral function of the F-distribution with $\left(K,1\right)$ degrees of freedom.

From Expression (28), it follows that there are essentially unequal distributions of statistics $\chi_{v,v}^{2}$ and a pair of statistics $\chi_{r,v}^{2}$, $\chi_{v,r}^{2}$ provided that r≠v. Thus, Expression (28) theoretically proves the correctness of Expressions (23) and (24) concerning the asymmetry of the value of information discrepancy, which is taken into account in the decision rule (22). This means that when the condition $\exists v,r\leq R:\rho_{v,r}>>\rho_{r,v}$ is satisfied, it is more appropriate to apply the decision rule (22) rather than (25) to make decisions about the recognition of language units in the speech signal parameterized in the paradigm of concept (8)–(10). This thesis will be tested in the experimental part of this article.

Assume that when recognizing the signal under study using the decision rule (25), the verdict was erroneously in favor of the hypothesis $W_{\mu }\left(X\right)$, not the hypothesis $W_{v}\left(X\right)$. Suppose also that when recognizing the same signal using decision rule (22), the verdict was made in favor of the hypothesis $W_{v}\left(X\right)$. The stated assumptions assume that according to Expressions (25) and (26), inequalities $\gamma_{v,v}\geq \gamma_{v,\mu }$ and $2\gamma_{v,v}\geq \gamma_{v,\mu }+\gamma_{\mu ,v}$ were fulfilled simultaneously, which is possible only if the condition $\gamma_{\mu ,v}>>\gamma_{v,\mu }$ is satisfied. Thus, an analytical indication of the erroneousness of the decision made under Rule (25) concerning the analyzed sample $X_{0}$ may be inequality of the form ${\bar{W}}_{\mu }\left(X\right):\gamma_{\mu ,0}>>\gamma_{0,\mu }$ or
(29)$${\bar{W}}_{\mu }\left(X\right):\frac{1+{\tilde{\gamma }}_{\mu ,0}}{1+{\tilde{\gamma }}_{0,\mu }}\geq c_{0},$$
where ${\tilde{\gamma }}_{0,\mu }=\frac{2\gamma_{0,\mu }}{n}$, ${\tilde{\gamma }}_{0,\mu }=\frac{2\gamma_{0,\mu }}{n}$ are the specific values of the solving statistics (23), (24), respectively; $c_{0}$ is the threshold value (minimum value of the asymmetry coefficient of the values (23) and (24) in Rule (22)), set depending on the maximum permissible error
$$\beta ≜P\left\{\frac{1+{\tilde{\gamma }}_{\mu ,0}}{1+{\tilde{\gamma }}_{0,\mu }}\geq c_{0}\left|W_{\mu }\right.\right\}\leq \beta_{0}.$$

Repeating the considerations that accompanied the transition from Expressions (26) to (28), we rewrite the defined expression to determine the probability β in terms of the F-distribution:(30)$$\begin{array}{c} \pi_{v\to \mu }≜P\left\{\frac{1+{\tilde{\gamma }}_{\mu ,0}}{1+{\tilde{\gamma }}_{0,\mu }}\geq c_{0}\left|W_{v}\right.\right\}=P\left\{\frac{1+{\tilde{\gamma }}_{v,\mu }}{1+{\tilde{\gamma }}_{\mu ,v}}\geq \frac{1}{c_{0}}\right\}\\ =P\left\{\frac{1+{\tilde{\gamma }}_{\mu ,\mu }}{1+{\tilde{\gamma }}_{0,0}}\geq c_{0}\right\}=P\left\{\frac{\chi_{\mu ,\mu }^{2}\left(K\right)}{\chi_{0,0}^{2}\left(K\right)}\geq c_{0}\right\}=1-\Phi_{K,K}\left(c_{0}\right)\leq \beta_{0}. \end{array}$$

Analyzing Expression (30), we obtain an equation $\min c_{0}=f_{K,K}\left(1-\beta_{0}\right)$, where $f_{K,K}\left(1-\beta_{0}\right)$ is the quantile of the F-distribution with $\left(K,K\right)$ degrees of freedom and the level of significance $1-\beta_{0}$. For example, for K=100 and $\beta_{0}=0.01$ from the tables for F-distribution, we have: $c_{0}\geq f_{100,100}\left(0,99\right)=1.59$.

Thus, Rule (29) allows us to estimate the probability of the event of marginal recognition of the correct result of the phoneme recognition procedure, employing the decision rule (25). The stochastic estimate of such an event is characterized by the expression
(31)$$\pi_{v\to \mu }=P\left\{\frac{\chi_{v,\mu }^{2}\left(1\right)}{\chi_{\mu ,v}^{2}\left(1\right)}\geq \frac{1+\rho_{v,\mu }}{c_{0}\left(1+\rho_{\mu ,v}\right)}\right\}=1-\Phi_{1,1}\left(\frac{1+\rho_{v,\mu }}{c_{0}\left(1+\rho_{\mu ,v}\right)}\right)$$
and is determined by the result of comparing the opposing elements $\rho_{r,v}$ and $\rho_{v,r}$ in the matrix $\left\|\rho_{r,v}\right\|$.

## 3. Results

We use Rule (11) based on the concept (8)–(10) to estimate the phonetic saturation of speech of persons in a team of 30 people. The personnel composition of this team was formed in a balanced way. It took into account such criteria as age (three age groups: 20–29, 30–39, 40–49 years), gender (male, female), higher education (university), native language (Ukrainian), and level of English language proficiency according to CEFR-B2. Each person listened to a phonogram of an 1800-character English-language journalistic text pronounced by a Google Translate service once. Subsequently, each person recounted the heard text for recording on a personalized digital phonogram lasting 3 min. The phonation of the retelling took place at the same tempo and timbre and with a clear fixation on language units. The phonograms were recorded using an AKG P420 microphone without an amplifier connected to a Creative Audigy Rx sound card integrated into a personal computer with a sampling frequency of 16 kHz. Each phonogram was saved in a .wav format file. For further analysis, the phonograms were split into segments of duration τ=5 ms (L=80 samples). Based on the analysis of the corresponding phonograms of retellings, individual phonetic alphabets $\left\{X_{r}\right\}$ were formed for each person, for which the centers of clusters of phonemes $\left\{x_{r}^{*}\right\}$ were determined by Expression (2). Two variants of the individual phonetic alphabet were formed for each person with hard and soft conditions of formation. These conditions were caused by the level of a mismatch $\Delta \rho =\left\{0,5;1,0\right\}$ for phonemes of the same name and their minimum duration $\Delta L=\left\{8L;4L\right\}$, $\tau =\left\{40;20\right\}$. The values of the autoregression coefficients $\left\{a_{r}\left(m\right)\right\}$, $\left\{a_{v}\left(m\right)\right\}$ required for the calculation of the information mismatch matrix $\left\|\rho_{r,v}\right\|$ were determined using the Berg–Levinson recurrent procedure with an unambiguously determined order of models p=20.

Figure 1 visualizes fragments of the resulting matrices for person №1, calculated with the selected hard (Figure 1a) and soft (Figure 1b) sets of formation conditions. The capacities of the phonetic alphabets were $R_{hard}^{1}=32$ and $R_{soft}^{1}=87$ language units, respectively. The minimum value of information discrepancy between phonemes was $\Delta \rho_{rv}^{R_{hard}^{1}}=0.324$.

Respectively, according to the decision rule (2), taking into account Expression (7), the value of the threshold $\rho_{0}=0.324$ is determined. Using the tables of the $\chi^{2}$-distribution for the number of degrees of freedom M=60, the probability of error of the first kind α=0.047 is determined. Then, according to Expression (13), the upper limit of the scattering of useful information of the phonation process for person №1 is equal to $\sup H\left(X\left|X^{\prime }\right.\right)=\alpha \log R=0.235$, and the upper limit of phonetic saturation of speech for person №1, according to Expression (11), is equal to $\sup I\left(X\left|X^{\prime }\right.\right)=\left(1-\alpha \right)\log R=4.765$.

Similar calculations were made for the rest of the persons in the team. For clarity of presentation, these results were averaged for each of the three age groups and visualized in Figure 2. In addition, for comparison, for persons from the first age group, the mismatch matrices were calculated with the selected soft set of formation conditions, and we performed all other computational operations described above. These results, referred to as «$1_{soft}^{AG}$», are also shown in Figure 2.

We investigate empirically the functionality of decision-making concepts generalized by solving Rules (22) and (25) in the task of the computational phonetic analysis of speech (statistical classification without a teacher in the concept (8)–(10) paradigm). The empirical material for the research was two phonograms with a recording of the same content of language material spoken by person №1. Phonograms were represented by samples $X_{0}$, $X_{r}$ of equal capacity M=120. First, the information mismatch matrix $\left\|\rho_{r,v}\right\|$ was calculated for four vowel phonemes of the person №1. The content of the matrix is visually presented in Figure 3. Allophones ${\left[u:\right]}_{1}$ and ${\left[u:\right]}_{2}$ represent person-specific dialects of pronunciation of the phoneme $\left[u:\right]$.

Further use of the data presented in Figure 3 will be demonstrated by the example. Consider the data from the matrix $\left\|\rho_{r,v}\right\|$ for a pair of phonemes $\left(\left[a:\right],{\left[u:\right]}_{1}\right)$. These data characterize the situation when the phoneme $\left[a:\right]$ is recognized as a phoneme ${\left[u:\right]}_{1}$. Figure 3 shows that $\rho \left(\left[a:\right],{\left[u:\right]}_{1}\right)=5,76$. The number of degrees of freedom for the F-distribution in expression (28) is assumed to be equal to K=M−p=100. If the decision on the result of phoneme recognition is made according to the solution rule (25), then $\left(K,1\right)=\left(100,1\right)$, and we have ${\tilde{\alpha }}_{v\to r}=1-\Phi_{K,1}\left\{1+\rho_{r,v}\right\}=1-\Phi_{100,1}\left\{5.76\right\}\approx 0.3$. If the decision on the result of phoneme recognition is made according to the decision rule (22), then we have $\max\left[\left(1+\rho_{vr}\right);\left(1+\rho_{rv}\right)\right]=\max\left[87.2;6.76\right]=6.76$. According to Expression (28), we have $\alpha_{v\to r}\leq 1-\Phi_{100,1}\left(5.76\right)\approx 0.12$. Thus, the probability of confusion when deciding on the result of phonetic analysis on the example of phonemes $\left[a:\right]$ and ${\left[u:\right]}_{1}$ using the solution rule (22) in comparison with Rule (25) is almost three times less. Calculations similar to the above were performed for all pairs of phonemes of different names in Figure 3. For all implementations, Rule (22) allowed us to obtain a lower estimate of the probability of confusion compared to Rule (25).

Let us complete this stage of research by calculating by means of Expression (31) the probability of the event of marginal recognition of the correct result of the phoneme recognition procedure using the decision rule (25): $\pi_{r\to v}=1-\Phi_{1,1}\left\{\frac{1+86.2}{1.59\left(1+5.76\right)}\right\}=1-\Phi_{1,1}\left(8,11\right)\approx 0.21$. It can be stated that the greater the asymmetry between the opposing elements of the matrix $\left\|\rho_{rv}\right\|$, the greater the value of probability $\pi_{r\to v}$.

We generalize the experimental section by verifying the models proposed in Section 2 in the paradigm of practical planning theory. We form certain sets of input influences (speech synals): $X^{k}=\left\{x_{1}^{k},x_{2}^{k},\ldots ,x_{n}^{k}\right\}$ and $X\bar{{}^{k}}=\left\{x_{1}^{\bar{k}},x_{2}^{\bar{k}},\ldots ,x_{m}^{\bar{k}}\right\}$. The system’s response to input effects from the set $X^{k}$ is predicted in the consept. Input influences from the set $X^{\bar{k}}$ are structurally identical to the generalized set $X^{k}$ but differ in values that may exceed the limits set up in the system’s design stage (extraneous noises, significant problems with diction, etc.) The system’s reaction to the input influence from the set $X^{\bar{k}}$ can be incorrect speech unit recognition. The numbers of elements in the sets $X^{k}$ and $X^{\bar{k}}$ are n=3000 and m=7000, respectively. Experiments were performed with fixation on the reaction of the system to the input influences from the sets $X^{k}$ and $X^{\bar{k}}$ (in the matrix form $B_{e}^{k}=\left(B_{ij}^{k}\right)$, $i=\bar{1,n}$, and $B_{e}^{\bar{k}}=\left(B_{ij}^{\bar{k}}\right)$, $i=\bar{1,m}$, respectively). We calculate for the *i*th input influence the variance of the implementation of the situation of the incorrect speech unit recognition: $s_{i}^{2}=M^{-1}\sum_{j=1}^{M}{\left(B_{ij}-{B^{\prime }}_{ij}\right)}^{2}$, where $B_{ij}$ is the state defined in the model; ${B^{\prime }}_{ij}$ is the actual state. We calculate the average value of the variance for all input influences: $s^{2}=N^{-1}\sum_{i=1}^{N}s_{i}^{2}$. Evaluation of the substantial deviations $s_{i}^{2}$ from $s^{2}$ Fisher’s criterion showed that all deviations do not exceed the tabular values, which confirms the adequacy of the proposed mathematical apparatus.

## 4. Discussion

The task of computational phonetic analysis of speech in the general case is reduced to a cyclically repeated procedure for estimating the deviation of the current segment of the studied speech signal from the etalons defined within a finite list of language units. The duration of the segments, by the sequence of which the output speech signal is presented, is selected based on the average duration of the studied language units; for example, for phonemes it is $\tau \in \left\{\bar{5,10}\right\}$ ms. In the paradigm of the Bayesian theory of pattern recognition, such a task is solved by testing stochastic hypotheses about the homogeneity of the distribution law of the speech signal. If the empirical distribution law can be reliably estimated by Gaussian approximation, then the above-mentioned task has an optimal solution. If the procedure of comparing the empirical segment with the etalon is trivial, then the question of determining the etalon for the language unit is a cornerstone. There is no generally accepted definition of the etalon of a language unit in the context of computational phonetic analysis of speech. A typical approach is to determine the desired etalon based on one of the variations of the method of expert assessments. However, this approach examines not so much the phonation of the language unit as the environment of distribution of signal and format of its presentation. In this context, the derivation of the task of computational phonetic analysis in the subject area of information theory allows us to consider the definition of the etalon in the absolute metric of the criterion of relative entropy, rather than in the relative metric, as implemented in analogues.

From the empirical results shown in Figure 2, we can draw conclusions about the representativeness of the proposed metric $\left\{H\left(X\left|X^{\prime }\right.\right);I\left(X\left|X^{\prime }\right.\right)\right\}$ for estimating the personified phonetic saturation of speech. It turned out that the highest phonetic saturation (11) is characterized by the speech of persons from the second age group. Of particular note are the data characterizing the phonetic saturation of speech of persons from the first age group, whose phonetic alphabets were determined by choosing a hard and soft set of formation conditions-$1_{hard}^{AG}$ and $1_{soft}^{AG}$, respectively. It is seen that the phonetic saturation of speech of persons from the first age group j$I_{hard}^{1AG}=4.765$, estimated based on phonetic alphabets $R_{hard}^{1AG}=32$, determined by choosing a hard set of formation conditions, was higher than the same indicator for the same group of persons $I_{soft}^{1AG}=4.304$, estimated based on phonetic alphabets $R_{soft}^{1AG}=87$, determined by choosing a soft set of formation conditions. This is without assuming that the capacity of the phonetic alphabet of the second variant $R_{soft}^{1AG}=87$ exceeds the capacity of the phonetic alphabet of the first variant $R_{hard}^{1AG}=32$ more than twice. This fact allows us to outline a promising direction for the investigation of the function $I\left(X,X^{\prime }\right)=f\left(R,\Delta \rho ,\Delta L\right)$, the extremum of which can potentially indicate the elements of the personalized phonetic alphabet, in which the individuality and informativeness of speech are most pronounced.

Based on the relative entropy function, Section 2.3 theoretically substantiates two error-detectable approaches to decision-making $W_{v}\left(X\right)$ in the task of computational phonetic analysis of speech (15) based on decision rules (22) or (25). The results of the computational experiment presented in Figure 3 convincingly prove the functionality of both of these approaches. Of particular importance is Expression (31) to estimate the reliability of a decision made based on Rule (25). Indeed, if the solution $W_{\mu }\left(X\right)$ is found to be erroneous according to Expression (29), then this fact, according to the provisions of the theory of experimental planning, will oblige the researcher to repeat the experiment according to Scheme (15) with all already rejected distribution alternatives, because the decision on their marginality is compromised. The result of such a re-experiment
(32)$${\tilde{\tilde{W}}}_{v}\left(X\right):{\tilde{\tilde{\lambda }}}_{v}\left(X\right)≜{\left.\gamma_{0,r}\right|}_{r=v\neq \mu }=\min,$$
determined on a reduced sample of alternatives with capacity R−1, together with the solution rules (25), (29), defines the entropy-based concept of detecting and correcting errors in the computational phonetic analysis of speech. The potential inherent in the proposed concept and the demonstrated results prove its superiority over such Bayesian concepts of decision-making using Euclidean-type mismatch metrics as the method of maximum likelihood and the method of the ideal observer.

Finally, it should be noted that the mathematical apparatus proposed in this article is proved to be adequate because it is based on the verified mathematical apparatus of information theory. This fact, as well as the rigor and reversibility of the analytical transformations carried out in the formalization of the corresponding metric, substantiate the adequacy of the mathematical apparatus presented in the article.

## 5. Conclusions

The study of a cornerstone object for modern linguistics, the process of speech and textual interpersonal communication, considering the size of the infosphere of the twenty-first century, is impossible without a thorough and purposeful involvement of information technology from other fields of knowledge, including computer science. Created as a result of relatively young science, computational linguistics aims to automatically analyze natural languages in all spectra of their implementation. From the long list of current tasks actively studied in the paradigm of computational linguistics, we mention the automation of compilation and linguistic processing of language corpora, the automated classification and abstracting of documents, the creation of accurate linguistic models of natural languages, ad the extraction of factual information from informal linguistic data. An effective, strictly formalized technology of computational phonetic analysis of linguistic information, especially speech information, is potentially the driving force behind the improvement of the results of solving these research tasks. This thesis is fully consistent with the content of the article, which proves the relevance of the presented scientific and applied results.

The proposed concept in this article (i.e., the mathematical model and methods) of computational phonetic analysis of speech defines the **scientific novelty** of the research. In the concept, in contrast with the existing methods, the task of addressing the multicriteria of the process of cognitive perception of speech by a person is strictly formally presented in the theoretical and analytical apparatus of information theory, pattern recognition theory and acoustic theory of speech formation. The obtained concept allows for determining accurately the phonetic alphabet of a person, taking into account their inherent dialect of speech and individual features of phonation, as well as detecting and correcting errors in the recognition of language units and reliably assessing the phonetic saturation of speech.

The proposed concept is represented by the decision rule (2), the decision statistics (4) and Expressions (7)–(9). The central element of the concept is the matrix of information mismatch $\left\|\rho_{r,v}\right\|$ of language units of the personalized phonetic alphabet of the speaker. The matrix $\left\|\rho_{r,v}\right\|$ is the basis for calculating the threshold $\rho_{0}$ for the implementation of computational phonetic analysis by Expression (7). With a known value of $\rho_{0}$, based on Expressions (2) and (5), the procedure of segmentation of the studied phonetic alphabet of a speaker into a set of phonemes, which with probability β=1−α are reliably recognized despite disturbing factors, and another set of phonemes, which with probability α are not reliably recognized. The use of Expressions (9) and (10) allows for clarifying the result of the segmentation procedure, taking into account the variability of the phonation of the studied language units, introduced by the individual features of the articulation of a particular speaker. 

The study of the results of computational phonetic analysis based on the function of relative entropy allowed for substantiating theoretically two detectable errors of the process of recognition of language units (15) based on solving Rules (22) and (25). Note the possibility, formalized by Expression (31), to estimate the reliability of the decision made based on Rule (25). If the solution is found to be compromised according to Expression (29), then with the help of a computational procedure with Scheme (15), it is possible to find erroneously recognized unreliable results of phonetic analysis and rehabilitate them. Thus, the **practical significance** of the proposed concept of computational phonetic analysis of speech lies in the fact that with its help, it is possible not only to single out phonetic units in speech signals, taking into account the individual features of speech formation, but also to detect and correct errors in the results of such an analysis.

The potential inherent in the proposed concept and the experimental results presented after Figure 3 prove its superiority over such Bayesian decision-making concepts using Euclidean-type mismatch metrics as the maximum likelihood method and the ideal observer method. The analysis of the studied speech signal carried out in the metric $\left\{H\left(X\left|X^{\prime }\right.\right);I\left(X\left|X^{\prime }\right.\right)\right\}$ based on the proposed concept allows for establishing reliably the phonetic saturation of speech, which objectively characterizes the environment of speech signal propagation and its source.

***Further research*** is planned to analyze the function $I\left(X,X^{\prime }\right)=f\left(R,\Delta \rho ,\Delta L\right)$, the extremum of which can potentially indicate the elements of the personalized phonetic alphabet, in which the individuality and informativeness of the speech of the person are most apparent. The authors hope that the results of such an investigation will increase the practical value of the proposed system of models for the precision phonetic analysis of speech [39].

## Figures and Tables

**Figure 1 entropy-24-01006-f001:**
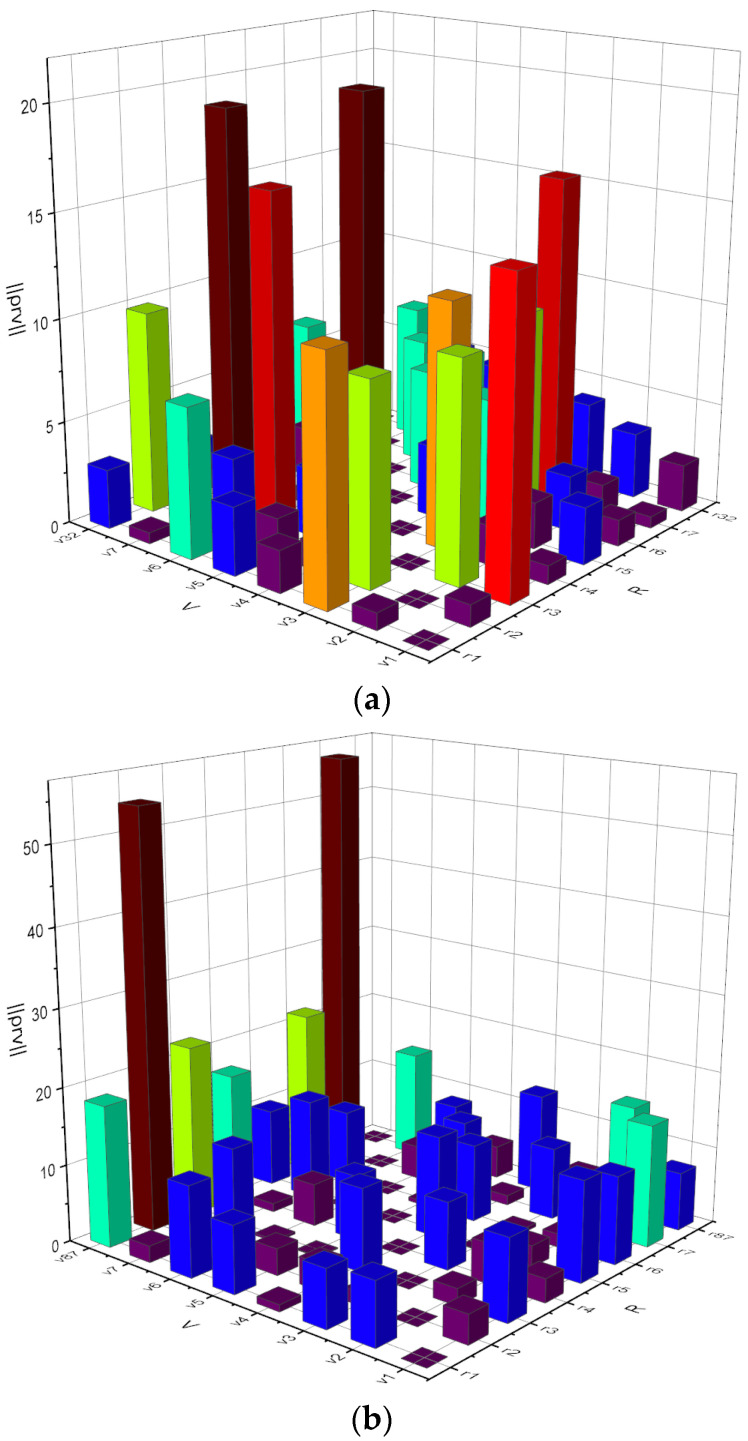
Visualization of fragments of information mismatch matrices $\left\|\rho_{r,v}\right\|$ for person №1, calculated with the selected hard (**a**) and soft (**b**) sets of formation conditions.

**Figure 2 entropy-24-01006-f002:**
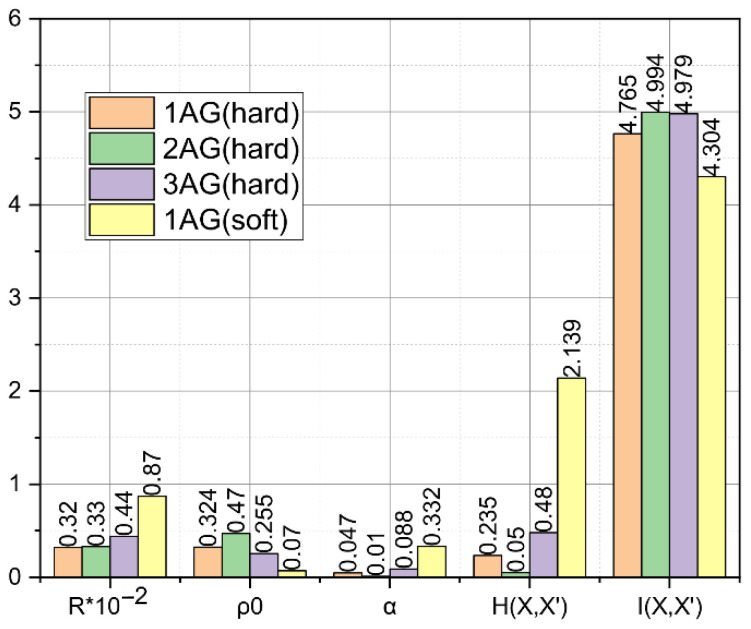
Estimation of phonetic saturation of personified speech (the format of the numbers is determined by the computing software used).

**Figure 3 entropy-24-01006-f003:**
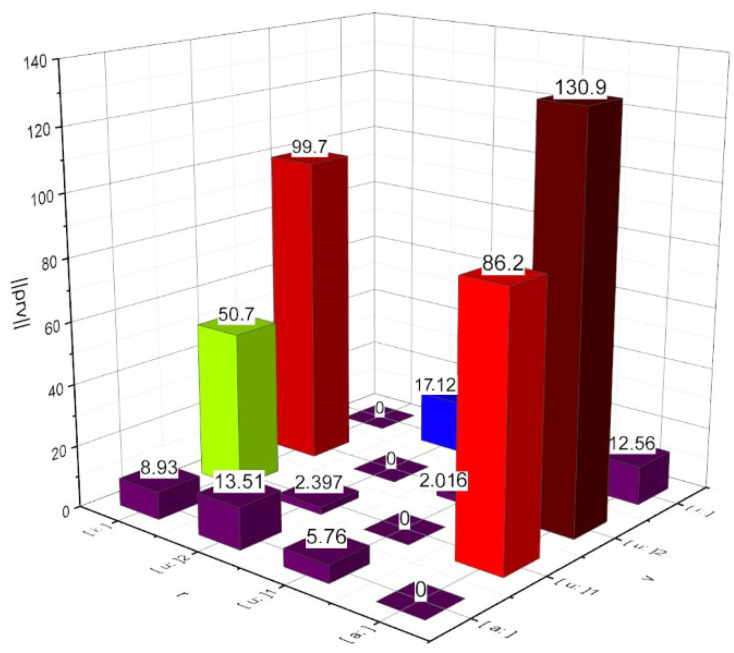
Visualization of a matrix $\left\|\rho_{r,v}\right\|$ calculated for instances of phonation of four phonemes by person №1 (the format of the numbers is determined by the computing software used).

## Data Availability

Most data are contained within the article. All the data available on request due to restrictions, e.g., privacy or ethical.

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
