# Peer review of "Entropy-Argumentative Concept of Computational Phonetic Analysis of Speech Taking into Account Dialect and Individuality of Phonation"

_entropy, 2022, doi:10.3390/e24071006_

Round 1

Reviewer 1 Report

The paper can be accepted after minor revision considering the remarks and recommendations:

- the scientific novelty and practical significance of research results should be described more clearly and concisely;

- schemes, algorithms of transformations which authors offer it is expedient to illustrate at least some drawings to improve clarity of presentation of materials of researches;

- expressions (25), page 13, and (32), page 22, are very close, and they need to be explained in more detail;

- authors write, page 4, “The highlights of the research are two entropy-argumentative concepts (of computational phonetic analysis of speech…, and detection and correction of errors of computational phonetic analysis of speech”. It would be importantly to specify term “concept”;

- it is not very clear how the obtained theoretical results and analytical models are verified;

- the discussion of research results (section 3) provided in section 4 is quite detailed, but it would be good to draw more general conclusions.

Author Response

Dear Colleague,

we sincerely thank you for the time devoted to our article and the excellent review. The recommendations you made are original, useful, and friendly. Writing answers was not a job for us, but a pleasure.

With wishes of happiness, health and creative success, Authors.

PS

Our answers are in the attached file.

Reviewer 2 Report

In general, the existing methods and contributions of this paper are not clearly differentiated. 

Authors should present a clear motivation and illustrate the problem clearly.

The statement "The concept of computational phonetic analysis of speech is proposed in the article for the first time" should be clarified. Is this  the contribution? The methods or analysis presented? Or the improvement of existing techniques?

Notation is not clear

Author Response

Dear Colleague,

we sincerely thank you for the time devoted to our article. The recommendations you made are strict but true. We took them into account in the revised version of the article. Please support our research.

With wishes of happiness, health and creative success, Authors.

PS

Our answers are in the attached file.
